

# A comparison of very short-lived halocarbon (VSLS) and DMS aircraft measurements in the Tropical West Pacific from CAST, ATTREX and CONTRAST

Stephen J. Andrews[1,2], Lucy J. Carpenter[1], Eric C. Apel[3], Elliot Atlas[4], Valeria Donets[4], James F. Hopkins[1], Rebecca S. Hornbrook[3], Alastair C. Lewis[1], Richard T. Lidster[1], Richard Lueb[3], Jamie Minaeian[1], Maria. Navarro[4], Shalini. Punjabi[1], Daniel Riemer[4], Susan Schauffler[3]

[1]Department of Chemistry, Wolfson Atmospheric Chemistry Laboratories, University of York, York, UK, YO10 5DD.
[2]Now at Center for Environmental Measurements and Analysis, National Institute for Environmental Studies, 16-2 Onogawa, Tsukuba 305-8506, Japan
[3]National Center for Atmospheric Research, Division of Atmospheric Chemistry, Boulder, CO 80307 USA
[4]University of Miami, Rosenstiel School for Marine & Atmospheric Science, Miami, FL 33149 USA

*Correspondence to*: Lucy Carpenter (lucy.carpenter@york.ac.uk)

**Abstract**

We present a comparison of aircraft measurements of halogenated very short-lived substances (VSLS) and dimethyl sulphide (DMS, $C_2H_6S$) from a co-ordinated campaign in Jan/Feb 2014 in the Tropical West Pacific. Measurements were made on the NASA Global Hawk, NCAR GV HIAPER and FAAM BAe146 using four separate GC-MS instruments operated by the University of Miami (UoM), the National Centre for Atmospheric Research (NCAR) and two from the University of York (UoY), respectively. The instruments were inter-calibrated during the campaign period using two gas standards on separate scales; a NOAA SX-3581 standard representative of clean low-hydrocarbon air, and an Essex canister prepared by UoM representative of coastal air, which was higher in VSLS and hydrocarbon content. UoY and NCAR use the NOAA scale/standard for VSLS calibration and UoM use a scale based on dilutions of primary standards calibrated by GC with FID and AED (atomic emission) detection. Analysis of the NOAA SX-3581 standard resulted in good agreement for $CH_2Cl_2$, $CHCl_3$, $CHBr_3$, $CH_2Br_2$, $CH_2BrCl$, $CHBrCl_2$, $CHBr_2Cl$, $CH_3I$ $CH_2ICl$ and $CH_2I_2$ (average RSD < 10 %). Agreement was in general slightly poorer for the UoM Essex canister with an average RSD of < 13 %. Analyses of $CHBrCl_2$ and $CHBr_3$ in this standard however showed significant variability, most likely due to co-eluting contaminant peaks, and a high concentration of $CHBr_3$, respectively. These issues highlight the importance of calibration at atmospherically relevant concentrations (~0.5 – 5 ppt for VSLS). The UoY in-situ GC-MS measurements on-board the BAe146 compare favourably with ambient data from NCAR and UoM, however the UoY whole air samples showed a negative bias for some lower volatility compounds. This systematic bias could be attributed to sample line losses. Considering their large spatial variability, DMS and $CH_3I$ displayed good cross-platform agreement without any sampling bias, likely due to their higher volatility. After a correction was performed based upon the UoY in–situ vs whole air data, all four instrument datasets show good agreement across a range of VSLS, with combined mean absolute percentage errors (MAPE) of the four platforms





throughout the vertical profiles ranging between 2.2 ($CH_2Br_2$) and 15 ($CH_3I$) % across a large geographic area of the Tropical West Pacific. This study shows that the international VSLS calibration scales and instrumental techniques discussed here are in generally good agreement (within ~10 % across a range of VSLS), but that losses in aircraft sampling lines can add a major source of uncertainty. Overall, the measurement uncertainty of bromocarbons during these campaigns is much

less than the uncertainty in the quantity of VSLS bromine estimated to reach the stratosphere of between 2-8 pptv (Carpenter et al., 2014).

# 1 Introduction

Halogenated very short-lived substances (VSLS) are defined as having atmospheric lifetimes shorter than tropospheric transport timescales, i.e. of around 6 months. VSLS (and their degradation products) are an important source of reactive

halogens to the lower stratosphere (Carpenter et al., 2014;Hossaini et al., 2015). The major source of brominated (e.g. $CHBr_3$ and $CH_2Br_2$) and iodinated (e.g. $CH_3I$) VSLS is the ocean, due to production by phytoplankton (Tokarczyk and Moore, 1994;Quack and Wallace, 2003), macro algae (Carpenter and Liss, 2000;Carpenter et al., 2000;Chance et al., 2009;Goodwin et al., 1997;Leedham et al., 2013;Schall et al., 1994;Sturges et al., 1993), bacteria and detritus (Asare et al., 2012;Hughes et al., 2008) and, for $CH_3I$, photochemically (Happell and Wallace, 1996;Richter and Wallace, 2004). However, major

uncertainties exist regarding the relative contribution of individual sources (Carpenter and Liss, 2000). Global distributions and controls of VSLS emissions are also not well known, exacerbated by large spatial variability in sea-air fluxes (Carpenter et al., 2005;Archer et al., 2007;Ziska et al., 2013;Stemmler et al., 2013;Orlikowska and Schulz-Bull, 2009). These are important considerations as deep convection in the tropics can rapidly transport VSLS to the upper troposphere/lower stratosphere (UTLS) and make a significant contribution to photochemical ozone depletion (von Glasow et al.,

2004;Salawitch et al., 2005;Yang et al., 2005;Montzka et al., 2011;Saiz-Lopez et al., 2012;Feng et al., 2007;Laube et al., 2008;Sinnhuber and Meul, 2015). Due to deep convection, the Tropical West Pacific supplies the largest source of stratospheric air (Fueglistaler et al., 2004;Bergman et al., 2012). VSLS are currently estimated to contribute 2-8 ppt of bromine to the stratosphere (Carpenter et al., 2014). Dimethyl sulfide (DMS) is also produced biogenically in the ocean and is the largest source of organic sulfur to the marine boundary layer (Cline and Bates, 1983;Nguyen et al., 1983;Andreae,

1986;Ferek et al., 1986). It affects the Earth's radiation budget and climate as an important aerosol and cloud condensation nuclei precursor.

Numerous global atmospheric transport models now include the VSLS $CHBr_3$ and $CH_2Br_2$ (Hossaini et al., 2016), driven by one of three ocean emission inventories by Liang (Liang et al., 2010), Ordonez (Ordonez et al., 2012) or Ziska (Ziska et al., 2013). Based on few measurements from limited geographical areas, these inventories are poorly constrained (Ashfold et al.,

2014) and contribute significantly to model uncertainties (Hossaini et al., 2013).

The Liang et al. (2010) and Ordonez et al. (2012) top-down inventories use simple latitudinal bands to define emissions, with equatorial and coastal enhancements informed by measurements. Major effort has been made to combine ocean and



atmospheric VSLS datasets and provide a bottom up (seawater-based) emission inventory (Ziska et al., 2013). For $CHBr_3$, the Ziska inventory results in the lowest mean absolute percentage error (MAPE) between the forecast and the measurements, compared to the other inventories (Hossaini et al., 2016).  However for $CH_2Br_2$, the Ziska et al. (2013) inventory tends to produce model overestimation of atmospheric mixing ratios, while Liang (2010), with the lowest emission

flux, performs favourably (Hossaini et al., 2016). One reason for poorer performance of the Ziska et al. (2013) inventory for $CH_2Br_2$ could be errors associated with measurements and variability between the compiled datasets, likely due to the lack of a common VSLS calibration scale.

Hossaini et al 2016 observed that, within the 12 models they compared for the TransCom-VSLS Model Inter-comparison Project, no single emission inventory was able to provide the best agreement between model and measurement at all surface

observation comparison locations.

In order for emissions inventories based upon collated measurement datasets to provide an accurate representation of surface VSLS distribution, it is imperative that datasets be properly inter-calibrated and compared such that their errors and variability are well characterized.

## 2 Experimental

### 2.1 Overview of campaigns

The dataset intercomparison consisted of three aircraft based campaigns: Co-ordinated Airborne Studies in the Tropics (CAST, Harris et al. (2016)), CONvective Transport of Active Species in the Tropics (CONTRAST, Pan et al. (2016)), and Airborne Tropical TRopopause Experiment (ATTREX). All 3 campaigns were carried out in the Tropical West Pacific in Jan/Feb 2014 and centred around the island of Guam (13.5°N, 144.8°E). No side-by-side comparison flights were carried

out, so we compare measurements on a statistical basis across a large geographic area of the west tropical Pacific where all 4 instruments sampled (130-165°E, 0-15°N, Figure 1, dashed purple boxes). The comparison region was limited to an altitude of 8 km, the ceiling for the BAe146 data. 1725 data points (158 AWAS, 490 TOGA, 629 WAS and 458 In-situ-GCMS) were sampled within this comparison region.

The comparison region contained predominantly open-ocean, well-mixed air masses, with some coastal input.  Elevated

surface levels of VSLS including $CHBr_3$ were observed near to coastal areas and islands; this is discussed on a case-by-case basis. Due to the relative homogeneity of open ocean VSLS emissions (e.g. Ziska et al., 2013), as shown by the relatively low variability of altitude-averaged data (see section 3.2), outside of the coastal areas we consider that a statistical comparison of data provides a good analysis of measurement comparability.

### 2.2 Methods

The comparison involved four different instruments using individual calibration gas standards and analysing air samples from three separate measurement platforms. These platforms were the UK Facility for Airborne Atmospheric Measurements





(FAAM) BAe 146 large research aircraft (CAST campaign), the National Science Foundation /National Centre for Atmospheric Research (NSF/NCAR) HIAPER Gulfstream GV (GV, CONTRAST campaign) and the National Aeronautics and Space Administration (NASA) Northrop Grumman Global Hawk (GH, ATTREX campaign).

On board the FAAM BAe 146, air samples were analysed by the UoY both by an in-situ GC-MS (coloured green in figures) and offline as Whole Air Samples (WAS, coloured blue in figures) by a separate GC-MS. Air samples captured by the NSF/NCAR GV and the NASA Global Hawk were analysed by the UoM and hereto referred by the collection apparatus name: Advanced Whole Air Sampler (AWAS, coloured red in figures). The fourth instrument was the NCAR in-situ GC-MS on board the GV, referred to as Trace Organic Gas Analyzer (TOGA, coloured black in figures).

### 2.2.1 Whole air sampler (WAS)

Samples were collected as described in Andrews et al. (2013) using evacuated 3 L SilcoCan canisters (Restek) sealed by pneumatically operated bellows valves (Swagelok, P/N SS-BNVVCR4-C). Air was drawn in through a forward facing air sampling pipe on the exterior of the aircraft and pressurised into the canisters using a metal bellows pump (Senior Aerospace PWSC 28823-7) to approximately 40 psig. The FAAM BAe146 has a much lower operating ceiling than the other aircraft but can profile down to ~15 m above sea level and sampled frequently in the marine boundary layer. Due to operating at tropical latitudes, this resulted in much higher than usually experienced humidity in the sample lines, pump and canisters. Analysis of the cylinders was carried out in the aircraft hangar, usually within 72 hours of collection. The stability of the measured VSLS in the WAS canisters was quantified over the course of one month and drift was found to be <-0.01 ppt per 24 hours for all species (Andrews, 2013). The positive pressure in the canisters was utilised for sample introduction into the instrument to avoid contamination from any potential leaks associated with reduced pressure sampling using a pump. Sample humidity was controlled using a -30 °C glass cold trap to remove water without loss of the analytes (Andrews 2015, Swan 2015). Two litres of sample was pre-concentrated onto the cooled (-30 °C) adsorbent trap (Tenax TA) of a thermal desorption unit (TDU, Markes Unity2-CIA-T) and desorbed with 2 mL /min Helium carrier at 250 °C onto a capillary column (Restek RTX502.2, 30 m, 0.25 mm I.D., 1.4 micron film thickness) of a gas chromatograph (Agilent 7890A) held at 40 °C for 3 min, then ramped at 25 °C min$^{-1}$ to 250 °C and held for 3 min. Analytes were detected using a Mass Selective Detector (MSD, Agilent 5977 Xtr source) with 19 selected ion windows monitoring a total of 46 ions, a qualifier and quantifier per analyte. The MSD source and quadrupole temperatures were 250 and 200 °C, respectively.

Calibration of the WAS instrument was carried out daily for VSLS using a NOAA (National Oceanic and Atmospheric Administration) calibration gas standard in a electropolished stainless steel canister (SX-3581 Essex Cyrogenics) filled in Oct 2013 and quantified using the NOAA 2003 scale for $CH_2Cl_2$, $CHCl_3$ and $CHBr_3$; NOAA 2004 scale for $CH_3I$ and $CH_2Br_2$; and a provisional scale for $CH_2BrCl$, $CHBrCl_2$, $CHBr_2Cl$, $CH_2ClI$ and $CH_2I_2$ the latter based on a limited number of standards in which mole fractions have remained consistent since 2009. NOAA SX-3581 contains an ambient air matrix with halocarbon concentrations enhanced to a few parts per trillion (Hall et al., 2014). For DMS, a custom standard was prepared



at the University of York containing an atmospherically relevant concentration of DMS diluted with nitrogen (BOC N6 grade). This was calibrated against a KRISS primary DMS standard.

Halocarbon concentrations in Essex cylinders were monitored at UoY and were found to be stable for >4 years after production, even for species such as $CH_2I_2$ which is often unstable in cylinders (Andrews, 2013). UoY have purchased three

NOAA calibration cylinders: SX-3570, SX-3576 and SX-3581 which all compared well to one another's quoted values (<4 %RSD) according to our analyses.

**2.2.2 In-situ Gas Chromatograph- Mass spectrometer (In-situ GC-MS)**

VSLS measurements were made in-flight using a thermal desorption (TD) GC-MS system mounted in the BAe146 cabin. Sample air was drawn from the same main sample line as for WAS and shared the same metal bellows pump within the

cabin. Air was diverted from the WAS pumping system before the sample line entered the aircraft hold and dried using a multi-core counter-current Nafion drier (Perma Pure PDseries). Samples were alternately pre-concentrated or analysed using dual, parallel adsorption traps (Tenax TA, Markes International TT-24/7), cooled to 0 °C. Analytes were refocused at the head of the column using liquid $CO_2$ prior to separation (10 m, 180 micron I.D., 1 micron film, Restek RTX502.2 column; 50 to 150 °C at 40 °C/min) by GC (Agilent 6850) and detection by electron impact MS selected ion monitoring (Agilent

5975C), calibrated pre-flight against the WAS gas standard (NOAA, SX-3581). Instrument temporal resolution, and associated sample integration period, was 5 min.

**2.2.3 Advanced Whole Air Sampler (AWAS)**

The AWAS sampler on the GV aircraft used custom-built electropolished stainless steel canister modules (1.3 L), with 12 canisters per module connected by a welded ¼" stainless steel manifold. Typically, 5 modules (60 canisters) were collected

20   per research flight. The canisters were sealed with pneumatically operated bellows valves (Swagelok, P/N SS-BNVVCR4-C, Swagelok, USA) and sample control was by computer command, either in an automatic or on-demand mode. The canisters were pressurised by two metal Bellows dual stage pumps (Model 28823-11, Senior Aerospace, Sharon, MA, USA), with a parallel input in the first two stages followed by serial connections in the second pump to provide final pressures of approximately 50 psia. Each module was cleaned with multiple, heated flushes of pure nitrogen, with a final addition of 8

torr of water vapour to passivate interior surfaces under dry (upper tropospheric) conditions. The inlet was an unheated stainless steel line connected to a HIMIL inlet (http://www.eol.ucar.edu/homes/dcrogers/Instruments/Inlets/) with the sample line at 90° relative to the airflow in the HIMIL. A heat exchanger was installed between the pump outlet and the sample inlet to remove bulk water from the sample flow. This inlet was the same as used on the HIPPO campaign for trace gas sampling. Sample flow through the system was dependent on altitude, and ranged from about 30 slpm to 5 splm. The GH

sampler consisted of modular sets of canisters of either 10, 8, or 6 cans per module. The 1.3 L canisters were custom built by Entech, Inc. to include a specially bent inlet tube to accommodate the module design. Each canister is coated with a proprietary silica based coating. The canisters were sealed with Parker Series 99 valves (P/N 099-0403-900, Parker Hannifin





Corp., Hollis, NJ, USA). Sample pumps were identical to the GV arrangement, but no heat exchanger was installed on the GH. An unheated, forward facing inlet with an exit for large particles and liquid water was installed on the underside of the GH aircraft. Sample canister preparation for the GH was the same as for the GV sampler.

Trace gas measurements from both aircraft were analysed on a single system. The system used a Markes Canister Interface
(CIA) and a Unity II system connected to an Agilent 5975 GC/MSD. The samples were dried in a -20°C stainless steel water trap, followed by a 2' section of nafion tubing (MD-050-24-FS-2; Perma Pure, Toms River, NJ) to further dry the sample. The sample was pre-concentrated on a Markes Ozone Precursor Trap (Markes UT17O3P-2S) held at -37°C. Sample size was 800 scc, with a controlled flow of 80 sccm. The sample was thermally desorbed at 300 °C for 6 minutes. The sample was split in the GC oven, with approximately 2/3 (525 cc) directed to a 30 m x 0.25 mm x 5 micron Alumina
PLOT column (HP-AL/S, Agilent Technologies) with a flame ionisation detector. A short (1m) section of GasPro column was added to the column end to facilitate separation of HFC-143a from ethyne. The remaining flow was sent to a 20 m x 0.2 mm x 1.12 micron DB-624 column (128-1324, Agilent Technologies). At the column exit, approximately 30% (83 cc) was split in an Agilent capillary splitter and directed to an electron capture detector. The remaining sample (about 192 cc) was sent to the MSD. The oven temperature program was  -20°C (3 min) 10 °C/min to 200°C (hold 4 min).

Calibration of the samples was done with between every 5 samples. The calibration gas was a whole air sample collected cryogenically at the Rosenstiel School of Marine and Atmospheric Sciences directly into an Essex 30 L cylinder. The calibration of the working standard was done by a series of dynamic dilutions of high concentration standards (Scott Specialty Gas) that had been previously measured by GC/AED (atomic emission detection) and GC/FID using NIST standards to verify carbon and halogen responses. The method has been described in Schauffler et al. (1999).

**2.2.4 Trace Organic Gas Analyzer (TOGA)**

VSLS and DMS measurements were made in-flight using a three-stage pre-concentrator coupled to a GC-MS system mounted in the GV cabin. Sample air was drawn through a heated electropolished stainless steel line by a metal bellows pump. The sampled air was not drawn through the pump but a subsample from the main inlet line was drawn at 25 mL per minute through a water trap (-25 °C) and into the enrichment trap (-130 °C) with a sample collection time of 35 seconds.
Following this, the enrichment trap was heated at 25 °C/s from -130 °C to 100 °C and the pre-concentrated sample was transferred with He carrier gas at 1 mL/min to the cryofocus trap which was cooled to -130 °C. The cryofocusing trap was then heated, also at 25 °C/s, from the cold set-point to +100 °C, in the presence of 1 mL flow of He carrier gas, thereby injecting the sample onto the custom designed gas chromatograph (GC). The GC was fitted with a Restek MXT-624 column (I.D. = 0.18 mm, length = 8 m). The initial GC oven temperature of 25 °C was held for 10 seconds followed by heating to
120 °C at a rate of 110 °C min$^{-1}$. The oven was then immediately cooled to prepare for the next sample. Sample processing time was two minutes. More details can be found in Apel et al. [2010]. The system was calibrated with NOAA standards SX-3515 and SX 3562 prior to deployment.




**2.2.5 MAPE calculation**

In order to compare platforms, the Mean Absolute Percentage Error (MAPE, %) was calculated for each instrument at each altitude bin. This shows how much each instrument has differed from the mean concentration of all data. For example, for the WAS 0-1000 m bin:

$$WAS\ MAPE_{0-1000} = \frac{100\%}{n} \sum_{dp=1}^{n} \left| \frac{C_{mean} - CWAS_{dp}}{C_{mean}} \right|$$

5                                                                            (1)

Where: $n$ = number of individual data points in the 1000 m altitude bin, $dp$ = individual instrument data point within the bin, $C_{mean}$ = mean concentration of all instrument data within the bin, $C_{dp}$ = WAS individual data point concentration within the bin.

**3 Results**

10   **3.1 Inter-calibration**

The three instruments were briefly compared via inter-calibration during the campaign. A 5 L SilcoCan canister (Restek), identical except for volume to those used for sampling aboard the FAAM BAe146, was evacuated and pressurised to 40 psig from the NOAA SX-3581 cylinder used to calibrate WAS. The same canister was used routinely to transfer calibration gas to the In-situ GCMS and for calibration during flights spanning multiple days. This was analysed by AWAS and TOGA within 15   72 hours of filling, in the same manner as a WAS sample. The analysis was performed 'blind' such that the concentrations were unknown to the analysts before providing their quantifications. A second inter-calibration gas provided by UoM was analysed by each institution. This was an Essex Cryogenics cylinder identical to NOAA SX-3581, filled cryogenically and analysed directly without decanting into a WAS canister. This sample was collected on the 2nd floor balcony of one of the campus buildings. Because the sample included traffic emissions as well as coastal marine emissions, the UoM cylinder 20   contained a more complex background matrix and significantly higher $CHBr_3$ concentration than SX-3581.

Considering the simplicity of the inter-calibration exercise, with just a few analyses from each of the two cylinders, the values reported by each institution are closely comparable (Figure 2). The average % relative standard deviation (RSD) between institutions for all analytes was < 10 % for the NOAA standard SX-3581. The average RSD for the UoM cylinder was slightly higher at < 13 %, mainly due to discrepancies in $CHBr_3$ and $CHBrCl_2$. The discrepancy in $CHBr_3$ is likely due 25   to the higher than ambient concentration in the UoM cylinder lying outside the calibrated range of WAS and TOGA and highlights the importance of calibration at ambient concentration ranges. The error encountered in $CHBrCl_2$ can be attributed to contamination from co-eluting chromatographic peaks. The UoY monitored ions with $m/z$ 127 and 129, which can suffer contamination issues but provide enhanced sensitivity when analysing 'clean' air such as from open ocean regions. It is


likely that TOGA instrument also suffers from contamination co-elution due to the fast chromatography employed. Air masses encountered during the CAST/CONTRAST/ATTREX campaigns were predominantly 'clean' open-ocean and more representative of the NOAA SX-3581.

## 3.2 Airborne data comparison

As the focus of the co-ordinated campaigns was to study the uplift of reactive halogens to the stratosphere, vertical profiles between platforms/instruments have been compared. This was performed by sorting the data into eight 1000 m altitude bins between sea-level and 8 km pressure height, a range sampled by all three instruments. Note, the NASA Global Hawk did not sample at these altitudes and therefore AWAS and TOGA are sampled from the same platform (GV).

We first compare $CHCl_3$, a predominately biogenic gas emitted mainly from the ocean (Laturnus et al., 2002), with a
moderately high background atmospheric mixing ratio (~10 ppt), and a relatively long atmospheric lifetime (150 days, (Carpenter et al., 2014)). All four raw (unadjusted) $CHCl_3$ datasets (figure 3) were comparable within bins to an average of 7 %RSD. To put this in perspective the average %RSD of the $CHCl_3$ data points within each bin is 12 % for WAS, 16 % for in-situ GC-MS, 15 % for AWAS and 25 % for TOGA.

This agreement between the datasets is very good considering that these measurements were taken on different times, days
and locations; from multiple sampling platforms and using different sampling techniques. However, the York WAS data is consistently lower than the in-situ-GCMS/AWAS/TOGA $CHCl_3$ data. This consistent offset is also apparent in other VSLS such as $CH_2Br_2$ (figure 4, bottom left), which is present at much lower concentration than $CHCl_3$ and remains fairly constant throughout the entire profile.

Considering the profiles of all measured species, it is apparent that the UoY WAS $CH_2Br_2$, $CHBr_3$, $CH_2BrCl$, $CHBrCl_2$,
$CHBr_2Cl$, $CH_2Cl_2$ and $CHCl_3$ data is consistently lower (19% +/-1 % throughout the vertical profile for $CH_2Br_2$) than the in-situ-GCMS, AWAS and TOGA measurements. We consider that the consistent bromo/chlorocarbon offset is not likely due to WAS sampling canister losses, which has been shown to be minimal (Andrews et al., 2013) and would likely have been seen during the inter-calibration when decanting SX-3581 into a WAS canister. One potential cause could be the sampling system aboard the FAAM BAe146. Due to the high humidity operating in the tropics, the WAS sampling lines often
contained a large quantity of water that was removed pre-flight. The WAS system samples from the main science air intake which runs the length of the aircraft. The sample is then pressurised into the WAS cylinders (stored in the aircraft hold) via unheated, stainless steel hoses.

In-situ VSLS analysis on-board the FAAM BAe146 was performed by a GC-MS calibrated by the same method and sharing the same sample inlet and pump as for the WAS system, with the exception that the in-situ-GCMS sampled much closer to
the outlet of the pump, before the long lines to WAS canisters in the aircraft hold where water collected pre-flight. The in-situ-GC-MS precision was not as good as for the WAS samples due to the fast chromatography employed, and profiles contained a larger data spread. However, the means and medians from the altitude binned data were consistently higher than for the WAS data and agreed well with AWAS and TOGA, supporting the theory that sampling line losses caused the offset





in WAS VSLS concentrations (Figure 4). We note that, unlike the bromo/chlorocarbons, $CH_3I$ and DMS did not show a consistent offset between in-situ and WAS data. This may have been due to their higher volatility and hence lower sampling losses compared to the rest of the substances analysed by UoY; these compounds also displayed a larger variability likely due to inhomogeneous ocean distributions and therefore comparisons between datasets are not straightforward.

The concurrent sampling of WAS and in-situ-GC-MS allowed a calculation of the WAS sampling offset for each affected compound. This was performed by averaging the offset at each altitude bin where the ratio of in-situ-GCMS:WAS was within 1σ of the average in-situ-GCMS:WAS ratio throughout the profile. This correction also improved agreement of the UoY WAS data with AWAS and TOGA, for example $CHCl_3$ improved from agreement within 7 %RSD across profile averages to 5 %RSD, $CHBr_3$ from 19 % to 7 % and $CH_2Br_2$ from 11 % to 3 %.

Above 5 km, the % RSD for many compounds increased with increasing altitude. This can be attributed to the fact that sampling of a range of relatively fresh to highly aged air masses occurred at these altitudes. The aged air masses, possibly entrained from higher altitudes or via long-range transport, can be identified by their characteristically high $O_3$ (Figure 6) or low water concentrations, associated with lower VSLS concentrations. BAe146 and GV transit flights were around 7 km and 12 km, respectively. The probability of each aircraft intercepting such air masses increases as a function of time spent at

those altitudes. Removing data where $O_3$ was greater than 50 ppb from each dataset removes this sampling bias from the higher altitude bins. All binned distribution plots here, with the exception of figure 3, have had the high $O_3$ data removed, including for the calculation of WAS offset.

In the $CHBr_3$ data especially, samples below 1 km show enhanced concentrations from localised emissions such as the atoll of Chuuk (figure 7), Palau island and Papua. WAS data showed the greatest enhancement as samples were taken from as low

as 87 m ASL and captured down-wind plumes from the islands. In-situ measurements such as the In-situ GCMS and TOGA collect an integrated sample (2 mins and 35 seconds respectively) that represents an average across the distance the aircraft has travelled in that period of about 3-6 km, whilst fill times for grab samples such as WAS and AWAS are usually around 5- 10 seconds (altitude dependant). Therefore grab sampling can under/overestimate average concentrations, especially at low sampling frequency, whereas in-situ measurements and high altitude grab samples will intrinsically average air masses.

This could account for the difference in $CHBr_3$ means between AWAS and TOGA and between WAS and In-situ GCMS at altitudes less than 1 km.

Combined vertical profiles for $CHBr_3$ and $CH_2Br_2$, the most atmospherically abundant bromocarbons (Figure 6), show a distinct "C" shaped profile with convective outflow visible at around 12-15,000 m. Low concentrations often coincided with enhanced $O_3$ concentration, consistent with air masses entrained from higher altitudes. In the case of the extremely low (<

0.5 ppt) $CH_2Br_2$ measured at ~13,000 m, the concentrations are lower than those measured at 18,000 m where the ozone concentration is similar and may suggest sampling of different air masses from long-range transport.



### 3.3 Comparison of MAPE

MAPE was calculated as described in section 2.2.5 and a summary of the vertical profiles of the average MAPE across all platforms, averaged across altitude bins, is shown in table 1. The average MAPE throughout the entire vertical profile for the 4 different instruments discussed here (In-situ-GCMS/WAS/AWAS/TOGA), compared to the mean observed

concentrations, was 7.7 % for $CHBr_3$ and 2.2 % for $CH_2Br_2$. Higher MAPEs were calculated for shorter-lived species including $CH_3I$ and DMS, where atmospheric variability is expected to make a large contribution to the spread of values measured across the different platforms.

In the TransCom-VSLS Model Inter-comparison Project, Hossaini et al. 2016 compared 12 models, using each models preferred emission inventory, with 7 recent aircraft campaigns including CAST. The model to measurement MAPE for all 7

campaigns was ≤ 35 % for $CHBr_3$ and < 20 % for $CH_2Br_2$. For most of the models studied by Hossaini et al., the model to ground-based observation MAPE over the latitudinal range ±20° was ~40 % for $CHBr_3$ and < 20 % for $CH_2Br_2$. Therefore, the differences between measurement techniques within this study are much smaller than the current differences between models and measurements.

### 4 Summary and discussion

We present a combined dataset of VSLS aircraft profiles in the West Tropical Pacific sampled by 3 different research groups on-board the NCAR GV and FAAM BAe146. Inter-calibration of standards showed that measurements agreed well (average RSD < 10 %) for the GC-MS analysis of $CH_2Cl_2$, $CHCl_3$, $CHBr_3$, $CH_2Br_2$, $CH_2BrCl$, $CHBrCl_2$, $CHBr_2Cl$, $CH_3I$ $CH_2ICl$ and $CH_2I_2$ in a NOAA SX-3581 standard containing ambient VSLS levels. Inter-calibration using an in-house standard cylinder containing super-ambient VSLS levels and contaminant peaks showed less good agreement for some compounds.

Comparison of the UoY WAS and in-situ GC-MS highlighted a negative bias in the WAS data which, by ruling out instrumental error and calibration scale differences via comparison and inter-calibration, we attribute to sampling losses in the BAe146 WAS sampling lines during the campaign. A future re-design could minimise such losses by locating the WAS cylinder instrument rack close to a window blank connected with short lengths of heated, inert tubing, allowing more direct sampling, analogous to the GV AWAS system.

Once the sampling losses are corrected for, agreement between averaged (by 1 km altitude bin) VSLS observations and the average of all (In-situ-GCMS, WAS, AWAS, TOGA) VSLS observations (the Mean Absolute Percentage Error, or MAPE) is very close, within 3 % for $CH_2Br_2$. Thus, it is apparent that spatial and temporal variability spanning months in this region of the West Tropical Pacific is fairly low. The BAe146 and GV never sampled the same air mass concurrently and yet the average MAPE for $CHBr_3$, usually a more spatially variable species, throughout all profiles is ~7 %, a value similar to the

average analytical uncertainty.

Our data is an encouraging result for global flux emission inventories as it suggests (1) relatively small analytical and scale errors across datasets, and (2) low spatial and temporal variability in tropical open ocean regions. Thus, it should be



perfectly valid for data to be collated and potentially extrapolated over large spatial areas, if systematic analytical biases can be eliminated. A step forward for the VSLS community would be to link their individual institutional calibration scales, both present data and historically, to one common scale, as in Hall et al. (2014). We have found (Andrews et al., 2013) that the stability of even the most reactive halocarbons ($CH_2ICl$, $CH_2I_2$) is excellent in Essex cryogenics air sampling canisters over

5   long time scales (> 3 years), when prepared using the NOAA method (Hall et al., 2014).

**Acknowledgements**

We acknowledge NERC for funding (NE/J00619X/1) and would like to thank the staff at FAAM, Directflight and Avalon Aero as well as the FAAM mission scientists for their work towards the success of the FAAM aircraft deployment in Guam.

10   We also acknowledge the support from the US National Science Foundation and NASA.

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

| Species | MAPE (%) | | | | | | | | |
|---|---|---|---|---|---|---|---|---|---|
| | 0-1 km | 1-2 km | 2-3 km | 3-4 km | 4-5 km | 5-6 km | 6-7 km | 7-8 km | 0-8 km |
| $CHBr_3$ | 10.5 | 9.7 | 7.3 | 4.0 | 8.6 | 7.0 | 6.8 | 7.6 | 7.7 |
| $CH_2Br_2$ | 3.1 | 2.4 | 1.6 | 1.0 | 2.3 | 2.8 | 1.6 | 2.5 | 2.2 |
| $CHCl_3$ | 4.4 | 2.3 | 1.6 | 3.4 | 3.6 | 2.0 | 5.2 | 3.4 | 3.2 |
| $CH_2Cl_2$ | 4.2 | 3.0 | 2.9 | 8.0 | 4.2 | 1.4 | 7.3 | 7.5 | 4.8 |
| $CH_2BrCl$ | 2.5 | 0.2 | 7.4 | 11.7 | 12.1 | 12.6 | 12.0 | 9.4 | 8.5 |
| $CHBrCl_2$ | 6.9 | 14.2 | 16.4 | 19.8 | 11.8 | 22.2 | 8.4 | 8.5 | 13.6 |
| $CHBr_2Cl$ | 3.2 | 13.1 | 7.4 | 9.6 | 12.8 | 14.1 | 11.4 | 10.9 | 10.3 |
| $CH_3I$ | 15.6 | 22.3 | 7.1 | 18.5 | 19.4 | 17.0 | 13.2 | 6.6 | 15.0 |
| DMS | 21.4 | 4.7 | 30.0 | 21.4 | 20.1 | 49.3 | 23.3 | 65.6 | 29.5 |



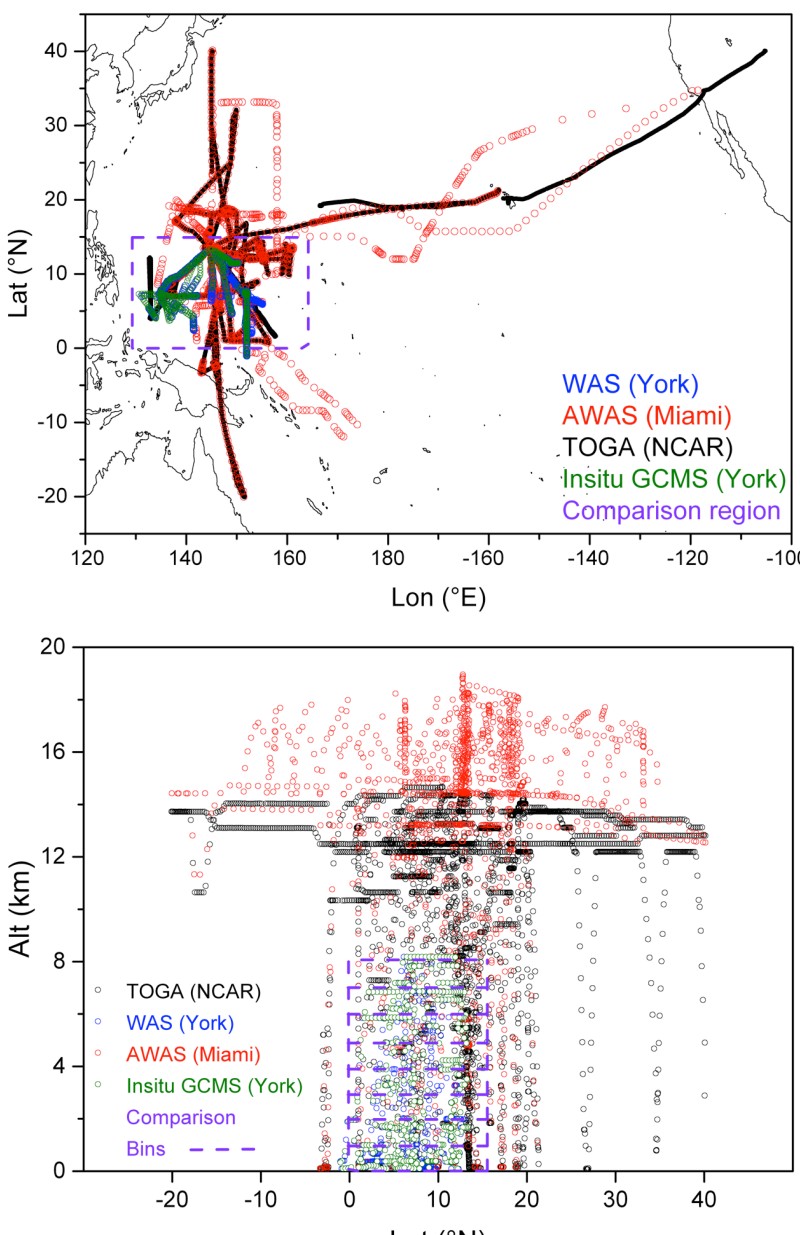

**Figure 1: Top: Spatial distribution of all CAST, CONTRAST and ATTREX VSLS measurements, coloured according to instrument, WAS (blue), AWAS (red), TOGA (black) and In-situ GCMS (green). Box (purple dashes) depicts region chosen for sample inter-comparison. Bottom: Vertical distribution of all CAST, CONTRAST and ATTREX VSLS measurements, coloured according to instrument. Purple dashed boxes show individual comparison bins (see section 3.2).**





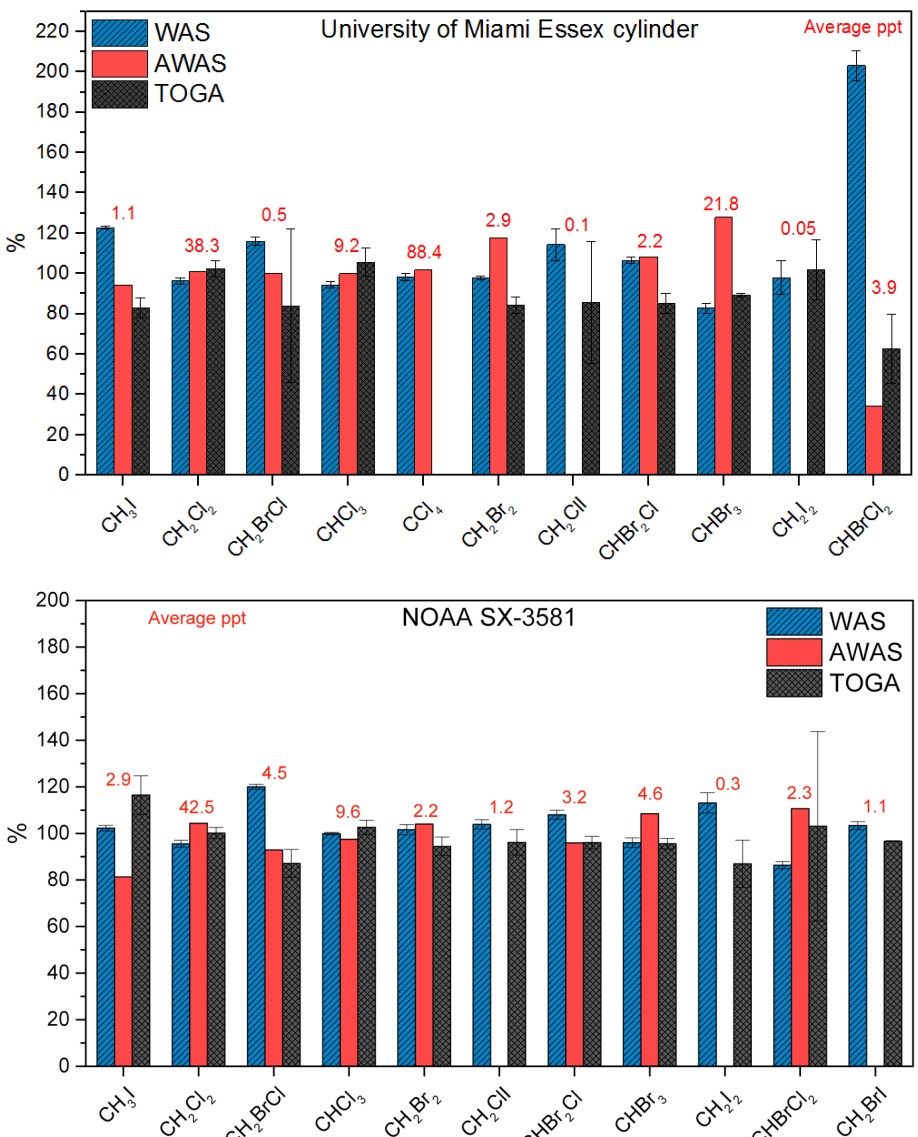

**Figure 2: Inter-calibration results from the two calibration gases, shown normalised to the average for each species. Values in red denote concentrations of analytes in the respective standards, whiskers show +- %RSD.**



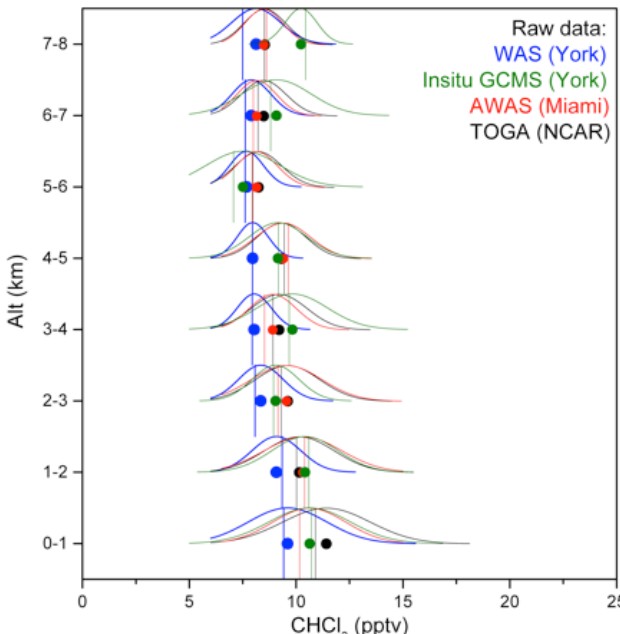

**Figure 3: Comparison of unadjusted CHCl₃ mixing ratios measured by all four instruments, averaged into 1000 m altitude bins. Circles and vertical lines are the means and medians of the binned data with the distribution represented by a normal distribution curve.**






**Figure 4: Vertical profile distribution bins showing a comparison of raw WAS data (purple), In-situ data (green) and WAS data adjusted to In-situ via the method described in the text (blue).**













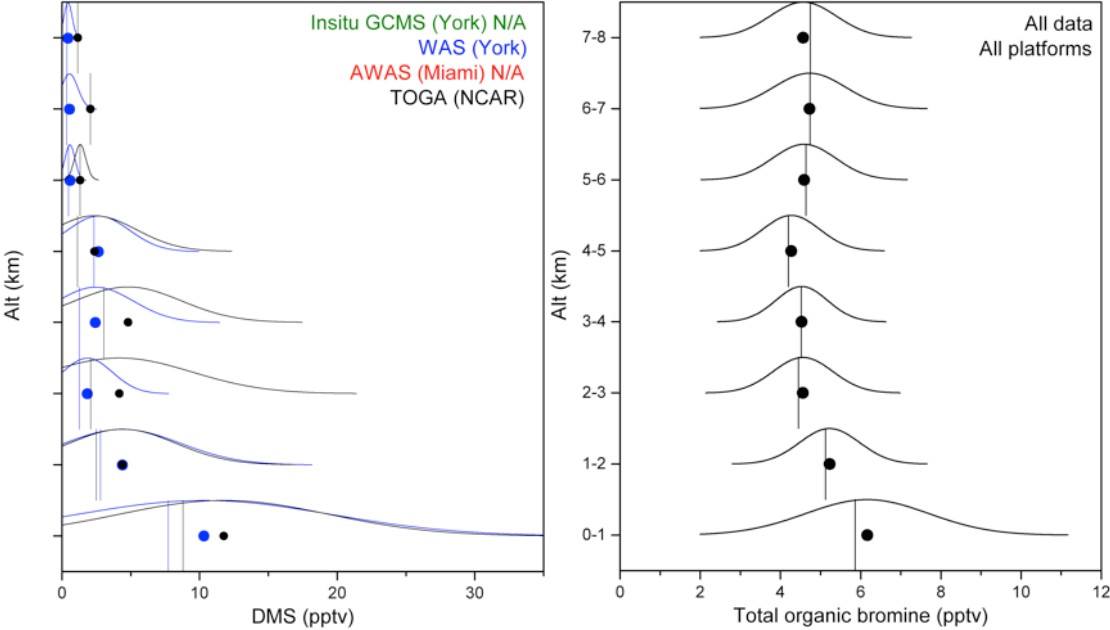

**Figure 5: Vertical profile distribution bins for all species. WAS data has been adjusted as described in the text. 'N/A' denotes cases where instruments have not analysed a compound. Bottom right: Total organic bromine vertical profile calculated by the addition of all contributing bromine atoms from organic halocarbons, averaged across all instruments/sampling platforms.**





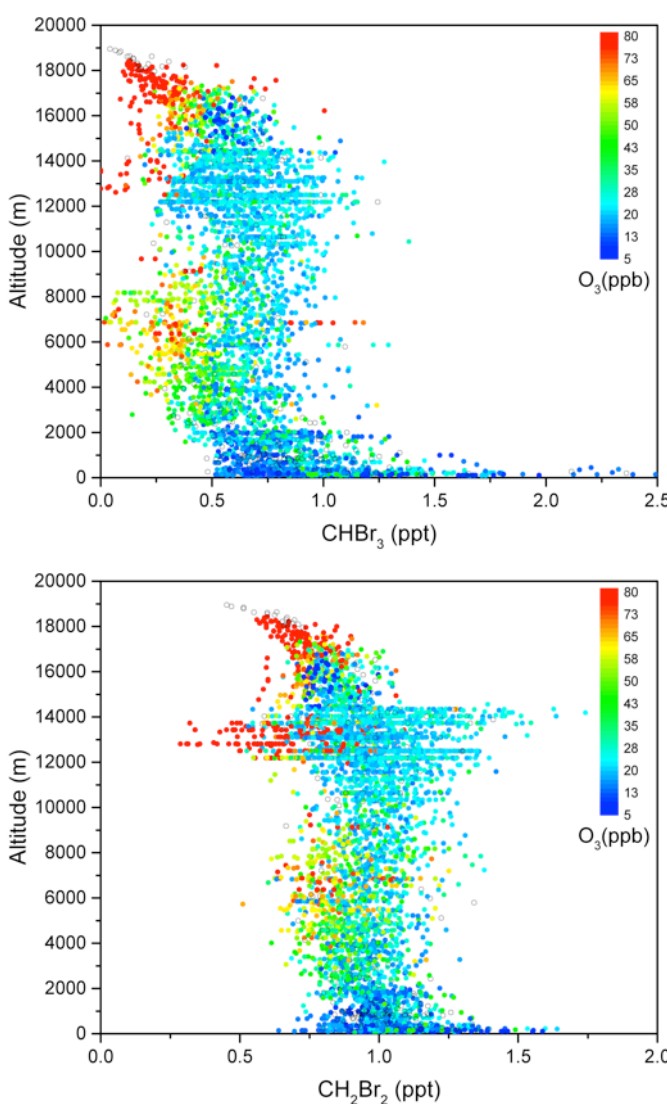

**Figure 6: Vertical profiles of CHBr₃ (top) and CH₂Br₂ (bottom) combined datasets, coloured by ozone concentration. Empty circles depict measurements with no corresponding ozone measurement.**




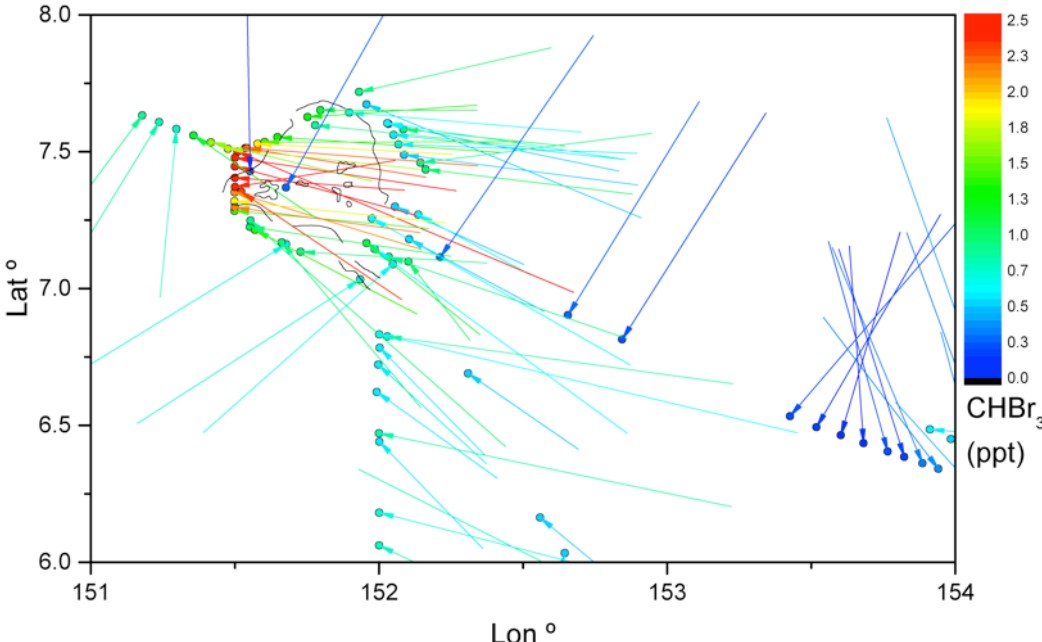

**Figure 7: Localised island influence from Chuuk Atoll. Circles depict WAS sampling sites, arrows show instantaneous wind direction and speed; both are coloured by CHBr₃ concentration.**

