# Peer review of "A comparison of very short-lived halocarbon (VSLS) and DMS aircraft measurements in the Tropical West Pacific from CAST, ATTREX and CONTRAST"

_Atmospheric Measurement Techniques, 2016_

## Referee Comment (RC1) · Anonymous Referee #1 · 10 May 2016

**Reviewer response: A comparison of very short-lived halocarbons (VSLS) and DMS aircraft measurements in the Tropical West Pacific from CAST, ATTREX and CONTRAST by Andrews et al.**
**AMTD doi:10.5194/amt-2016-94**

This is a well thought out and concisely written paper that I recommend by published with minor corrections.

(1) There is no standard intercomparison for DMS (Fig. 2 etc.) – this should at least be discussed. DMS was also only compared between two instruments. Compared to the extent of halocarbon data in this paper it is harder to draw conclusions on a comparison of DMS measurements between groups. I feel this paper needs to be clearer in the Abstract/Intro that DMS is a 'sideline' measurement in this paper, compared to the halocarbons.

(2) I feel the authors could improve the colour selections in Figs. 3-5. It is difficult to distinguish between the blues/purples (e.g. Fig. 4) and a red/green combination can also cause problems.

(3) Why was the UofY in-situ-GCMS not included in the intercomparison (Fig. 2)? This should be discussed.

(4) The authors do a good job of defining each measurement and associating it an acronym that is used throughout the paper. However, I feel the introduction to these acronyms needs some tidying up. On p.3 line 22 the acronyms 'AWAS', 'WAS' and 'in-situ-GCMS' are introduced but without explanation. A few lines later (p.4, line 4) the abbreviation 'UoY' is introduced without being explained in full (if one considers the abstract separately).

It may be worth having a table at the start of the method section that out all the information, e.g.:

| Platform | Institution (abbreviation) | Instrument | Instrument description | Sampling altitude range | Number of samples used |
|---|---|---|---|---|---|
| UK FAAM BAe 146 | University of York (UoY) | In-situ GCMS | Section 2.2.2 | | |
| … | … | Whole Air Sampler – WAS | Section 2.2.1 | | |
| … | … | AWAS | | | 158 |

(5) Page 5, line 1 – 'atmospherically relevant concentration' – can you provide specifics? You mention the benefits of using a calibration gas at ambient concentration ranges (e.g. p.7, line 26) so it would be helpful to give the concentration in all of the standards used by the groups involved in this comparison and how that compares to the ambient concentration range.

(6) You mention that water built up in the long sampling lines to the WAS canisters pre-flight (p.8, line 30) but also that this was removed pre-flight (p.8, line 25) – so what is the relevance of the water? Are you suggesting water remained in the line and contributed to losses? Or, as the water was removed, are there other aspects of the sampling line that may have led to losses? If it is water, can this not be tested with your data? For example, does the discrepancy increase over flight time – which would suggest something building up in the sampling line over time.

(7) Finally, there are a few typesetting issues. For example a lack of a space between multiple references (e.g. p.2, line 10) and a lack of capital letters when referring to specific tables/figures (e.g. p.8, line 9). However, I imagine these will be ironed out during processing and proofing for publication.

---

## Referee Comment (RC2) · Anonymous Referee #2 · 2 Jun 2016

This is a review of the paper titled "A comparison of very short-lived halocarbon (VSLS) and DMS aircraft measurements in the Tropical West Pacific from CAST, ATTREX and CONTRAST" by Andrews et al. Overall, the paper is well written and has completed a rigorous data analysis. I like the method of displaying the distributions as a function of altitude for each gas. I have some criticism that needs to be address: (1). Why are many of the sampling canisters treated with silicon compounds and the standards are not? (2). Why don't you test your theory in the lab of lost in the long tubing on the aircraft, rather than speculate. (3). I would like to see more discussion of Table-1 results as a function of altitude. Is the variation between datasets a result of altitude

[Figure]

**[AMTD](...)**

Interactive
comment

or reactivity/stability of gas or concentration? (4). Minor: grammar errors, data are plural. Do a search and replace over the whole manuscript. (5). It was very hard to see the individual reference citations without hanging indents. Perhaps this will be fixed by the journal in final form. I feel that the format of the journal, AMT, is appropriate for this article. Once the major points (1-3) are addressed, I would reconsider the revised manuscript for publication.

———————————————

---

## Referee Comment (RC3) · Anonymous Referee #3 · 7 Jun 2016

The paper by Andrews et al. presents an intercomparison of airborne VSLS measurements. The paper is well written, the message is clear, the conclusions are well supported.

Next to a few minor points, I have one major comment, which I would like the authors to clarify.

Major point: The "average VSLS MAPE" for $CH_2Br_2$ is quoted at 3%. This values is however only achieved after correcting the loss in the sampling lines. This loss is calculated through comparison with other observations (in-situ) GC. Correcting one value

with respect to another and then averaging these two values is a circular argument. The MAPE should be calculated without the correction of sampling line errors. These offsets can usually (if only a single data set is available) not be derived, so they must be included in the assessment of uncertainty.

Minor points:

p. 2. l. 17. can the drifts be specified in relative units?

p.2. l. 27: This is not a calibration of the WAS instruments but of the GC analyzing the WAS.

Section 2.2.2. Is there no in-flight calibration of the GC-MS? If so, please specify how detector drift is taken into account. Or is the pre-flight calibration only to ensure that standards between both instruments re consistent?

p.8. l. 30.: can the precisions of the of the individual instruments be specified? E.g. Sala et al. (ACP, 2014) present in-situ airborne GC-MS measurements of VSLS with partly very good precision.

p. 9. l.7.: why is tis procedure limited to 1sigma around the average? Should sampling line offsets not be altitude dependent, as the flow usually decreases with altitude? Also humidity changes with altitude.

p. 10 l. 5 and 27: I think that the way these numbers are calculated contains a circular argument (see major comment above).

---

## Author Comment (AC1) · 25 Jul 2016

We thank the three reviewers for taking the time to read our manuscript and providing constructive comments.

Response to reviewer 1:

(1) There is no standard inter-comparison for DMS (Fig. 2 etc.) – this should at least be discussed. DMS was also only compared between two instruments. Compared to the extent of halocarbon data in this paper it is harder to draw conclusions on a comparison of DMS measurements between groups. I feel this paper needs to be clearer in the

[Figure]

Abstract/Intro that DMS is a 'sideline' measurement in this paper, compared to the halocarbons.

Response: We agree that the lack of inter-comparison of DMS is disappointing but a comparable data set by two aircraft using different instruments, in-situ and canister sampling, is very encouraging and somewhat surprising for such a spatially variable and reactive species. Although DMS could be considered 'side line' to the extent of halocarbon data, if considered as an individual measurement the many hundred in-situ (GV) and canister samples (WAS) measured from two separate platforms during an extensive, co-ordinated flight campaign would be a worthy dataset in it's own right. This is a tricky compound to analyse, hence it only being reported by two of the instruments.

Changes to manuscript: We don't feel that the title is misleading in saying that DMS is compared across platforms but perhaps your views on our presentation of DMS data show that we have not focussed enough on the DMS measurements within the discussion of both comparison between instruments and the atmospheric distribution. This will be amended in the revised paper, in addition to mentioning the lack of inter-comparison.

(2) I feel the authors could improve the colour selections in Figs. 3-5. It is difficult to distinguish between the blues/purples (e.g. Fig. 4) and a red/green combination can also cause problems.

Response: We will try a different colour combination for the blues/purples. We wanted to keep the colours consistent throughout the paper as not to confuse the reader and the complex nature of the plots doesn't suit dashed lines etc. There are only so many colour combinations that can be used to produce a high contrast plot and we tried many before settling on the red/blue/green/black combo, which provides the best viewing for the majority of readers. Colour blindness is not always green/red. Using vivid high contrast colours is often the most distinguishable; hence we will look to change the pastel purple.

Changes to manuscript: Look at changing the blue/purple colour combination as not high enough contrast between them

(3) Why was the UofY in-situ-GCMS not included in the inter-comparison (Fig. 2)? This should be discussed.

Response: This was a combination of the aircrafts, instruments and standards being available at different times throughout the campaign, restrictions on movement of canisters and canister certification on aircraft. Aircrafts and instruments were located on both commercial and military airstrips and access issues hampered inter-comparison.

(4) The authors do a good job of defining each measurement and associating it an acronym that is used throughout the paper. However, I feel the introduction to these acronyms needs some tidying up. On p.3 line 22 the acronyms 'AWAS', 'WAS' and 'insitu-GCMS' are introduced but without explanation. A few lines later (p.4, line 4) the abbreviation 'UoY' is introduced without being explained in full (if one considers the abstract separately). It may be worth having a table at the start of the method section that out all the information,

Response: Indeed the number of aircraft and instruments make defining each quite complicated. We have tried to ensure that we are as consistent as possible with acronyms, colours, etc. throughout the paper and appreciate any help in making this easily understandable for the reader.

Changes to manuscript: A table may work well for this and will be explored.

(5) Page 5, line 1 – 'atmospherically relevant concentration' – can you provide specifics? You mention the benefits of using a calibration gas at ambient concentration ranges (e.g. p.7, line 26) so it would be helpful to give the concentration in all of the standards used by the groups involved in this comparison and how that compares to the ambient concentration range.

Response: The concentration for each analyte is shown in red on each inter-calibration

bar chart and described in the chart description. The atmospheric concentrations are spatially variable but the measurement plots shown in the paper clearly show the range of concentrations encountered throughout vertical profiles in the measurement region.

Changes to manuscript: The text will be amended to make clear that the calibration range should be similar to the concentration range encountered in air samples, shown here in the measurements.

(6) You mention that water built up in the long sampling lines to the WAS canisters preflight (p.8, line 30) but also that this was removed pre-flight (p.8, line 25) – so what is the relevance of the water? Are you suggesting water remained in the line and contributed to losses? Or, as the water was removed, are there other aspects of the sampling line that may have led to losses? If it is water, can this not be tested with your data? For example, does the discrepancy increase over flight time – which would suggest something building up in the sampling line over time.

Response: Due to the convective nature of the operating region, the aircraft was constantly exposed to very heavy rain and could not always be parked in a hangar. The water in the sampling lines could only be removed briefly pre-flight as it required powering the aircraft to pressurise the pumps and flush the lines. Some water certainly remained in the lines after this process and would enter the canisters during filling. Also, the primary section of the main sample inlet is not pumped and is a 'ram-air' style inlet that cannot be flushed whilst the aircraft is stationary. Its description will be added to the text and that this was flushed during flight before pumps were powered on to pressurise this stream into the cylinders. The In-situ GC_MS also used this line and did not show the same analyte losses so it is likely that they occurred as the sample line descended into the aircraft hold where the cylinders are stored. Water could accumulate in this region and temperatures could become cold. A prediction of wall losses in this line would be consistent with the observation that the higher volatility species showed no losses.

Changes to manuscript: Clarification of wording in manuscript

(7) Finally, there are a few typesetting issues. For example a lack of a space between multiple references (e.g. p.2, line 10) and a lack of capital letters when referring to specific tables/figures (e.g. p.8, line 9). However, I imagine these will be ironed out during processing and proofing for publication.

Changes to manuscript: minor grammatical errors and formatting will be fixed in final form as will be submitted in the journal latex template.

Response to reviewer 2:

This is a review of the paper titled "A comparison of very short-lived halocarbon (VSLS) and DMS aircraft measurements in the Tropical West Pacific from CAST, ATTREX and CONTRAST" by Andrews et al. Overall, the paper is well written and has completed a rigorous data analysis. I like the method of displaying the distributions as a function of altitude for each gas. I have some criticism that needs to be address: (1). Why are many of the sampling canisters treated with silicon compounds and the standards are not?

Response: The standard cylinders are electropolished stainless steel for halocarbons and Air liquide Experis cylinders for DMS. The long-term (years) stability of analytes has been verified in these. Different aircraft use different cylinders for many different reasons. Some are commercially available and some are custom made. Different aircraft have different payloads and size requirements. Each group has tested the stability of analytes in their sampling cylinders. In the case of the WAS samples, stability has been monitored for short-term (weeks) in the coated Restek Silco-cans and samples are usually analysed within a few days.

Changes to manuscript: may clarify wording in description of instruments

(2). Why don't you test your theory in the lab of lost in the long tubing on the aircraft, rather than speculate.

Response: This would be nice to test; however the sampling lines are fixed within the aircraft and cannot be removed. Trying to simulate something in the lab with the same conditions experienced would have too many assumptions and likely much less representative of the real sampling lines. We essentially had an instrument at the start of the sample lines and canisters at the end. The losses must have occurred within the lines or the cylinders. We are able to test the cylinders and find no losses. We are unable to test the lines and so must assume this is the source of the losses. This may be confined to data from this campaign only, or it may affect future campaigns. Either way, the finding should be investigated if further halocarbon measurements are to be made with this sampling system.

Changes to manuscript: text will be changed to clarify the sampling line description and the reasoning for suspecting losses in this section of line.

(3). I would like to see more discussion of Table-1 results as a function of altitude. Is the variation between datasets a result of altitude or reactivity/stability of gas or concentration?

Response: Table 1 MAPEs are a function of altitude as all of these gases have an ocean source. The aircraft are not sampling exactly the same air and as the emissions are oceanic, the spatial variability is bound to be greater at lower altitudes. As the air is convected and mixed, it becomes more homogeneous and thus the variability decreases.

Changes to manuscript: This is discussed in the text but not explicitly with regard to table 1. We will look to make this clearer in the revised manuscript.

(4). Minor: grammar errors, data are plural. Do a search and replace over the whole manuscript. (5). It was very hard to see the individual reference citations without hanging indents. Perhaps this will be fixed by the journal in final form. I feel that the format of the journal, AMT, is appropriate for this article. Once the major points (1-3) are addressed, I would reconsider the revised manuscript for publication.

[Figure]

Changes to manuscript: minor grammatical errors and formatting will be fixed in final form as will be submitted in the journal latex template.

Response to reviewer 3:

The paper by Andrews et al. presents an inter-comparison of airborne VSLS measurements. The paper is well written, the message is clear, the conclusions are well supported. Next to a few minor points, I have one major comment, which I would like the authors to clarify. Major point: The "average VSLS MAPE" for CH2Br2 is quoted at 3%. This values is however only achieved after correcting the loss in the sampling lines. This loss is calculated through comparison with other observations (in-situ) GC. Correcting one value with respect to another and then averaging these two values is a circular argument. The MAPE should be calculated without the correction of sampling line errors. These offsets can usually (if only a single data set is available) not be derived, so they must be included in the assessment of uncertainty.

Response: We accept this valid comment and will provide uncorrected MAPE in addition to corrected MAPE. We feel that a corrected MAPE is useful as it shows the variation in the concentrations better, without the added error of the losses in one system. An uncorrected MAPE will be useful as it is a better representation of the spread of sampling uncertainty.

Changes to manuscript: We will provide uncorrected MAPE in addition to corrected MAPE.

Minor points: p. 2. l. 17. can the drifts be specified in relative units? p.2. l. 27: This is not a calibration of the WAS instruments but of the GC analyzing the WAS.

Changes to manuscript: Noted and will be changed to clarify

Section 2.2.2. Is there no in-flight calibration of the GC-MS? If so, please specify how detector drift is taken into account. Or is the pre-flight calibration only to ensure that standards between both instruments re consistent?

Response: No in-flight calibration was available due to restrictions on cylinder types on the aircraft. The instrument was calibrated before and after flights and interpolated with a linearity check against measured, well mixed, long-lived halogenated atmospheric gases.

Changes to manuscript: This will be clarified in the instrument description.

p.8. l. 30.: can the precisions of the of the individual instruments be specified? E.g. Sala et al. (ACP, 2014) present in-situ airborne GC-MS measurements of VSLS with partly very good precision.

Response: The precisions are shown by the %RSD error bars in the inter-comparison-figure 2

p. 9. l.7.: why is tis procedure limited to 1sigma around the average? Should sampling line offsets not be altitude dependent, as the flow usually decreases with altitude? Also humidity changes with altitude.

Response: The data was adjusted with altitude dependence, this is perhaps not clear enough in the paper. The WAS data is adjusted to the In-situ data for each altitude bin, therefore if there is an altitude dependence in the ration of WAS/Insitu then this would be accounted for.

p. 10 l. 5 and 27: I think that the way these numbers are calculated contains a circular argument (see major comment above).